# Her-2 Targeted Therapy in Advanced Urothelial Cancer: From Monoclonal Antibodies to Antibody-Drug Conjugates

**DOI:** 10.3390/ijms232012659

**Published:** 2022-10-21

**Authors:** Víctor Albarrán, Diana Isabel Rosero, Jesús Chamorro, Javier Pozas, María San Román, Ana María Barrill, Víctor Alía, Pilar Sotoca, Patricia Guerrero, Juan Carlos Calvo, Inmaculada Orejana, Patricia Pérez de Aguado, Pablo Gajate

**Affiliations:** Department of Medical Oncology, Ramon y Cajal University Hospital, 28034 Madrid, Spain

**Keywords:** urothelial cancer, Her-2, targeted therapy, monoclonal antibody, tyrosine-kinase inhibitor, antibody-drug conjugate

## Abstract

Metastatic urothelial cancer, associated with a poor prognosis, is still major cause of cancer-related death, with scarce options of effective treatment after progression to platinum-based chemotherapy and immunotherapy. The human epithelial growth factor receptor 2 (Her-2) has been identified as a new therapeutic target in medical oncology. However, despite the encouraging results in breast and gastric cancers, clinical trials with anti-Her-2 monoclonal antibodies and tyrosine-kinase inhibitors have shown limited efficacy of this strategy in urothelial tumors. Notably, more favorable data have been recently shown that antibody-drug conjugates are currently emerging as a novel promising approach for Her-2 targeted therapy in advanced urothelial cancer.

## 1. Introduction and Background

Urothelial bladder cancer (UC) is among the top ten most frequent cancer types in the world and accounts for around 3% of new malignant diagnosis, with a higher incidence in men (3:1) and an increasing prevalence in Europe [1]. Tobacco smoking is the main risk factor, with a 50–60% population-attributable risk of UC [2]. One-fourth of the patients present with muscle-invasive bladder cancer (MIBC) at diagnosis, and up to 50% of patients with high-risk non-MIBC progress to invasive disease, in which radical cystectomy remains the gold standard of treatment, even for patients above 85 years of age [3]. Unfortunately, nearly half of the patients have a metastatic relapse after radical surgery [4]. Metastatic UC (mUC) remains a remarkable cause of cancer-related mortality, with a 5-year survival rate of 6% [5].

Platinum-based chemotherapy (ChT) is the frontline treatment for advanced or metastatic disease, followed by maintenance avelumab for tumors which have not progressed on ChT. Pembrolizumab or atezolizumab are also a possibility for cisplatin-ineligible patients with a programmed death-ligand (PD-L1) positive determination. Immune checkpoint inhibitors (ICIs) are standard in platinum-refractory disease, though other ChT regimes may be used for patients in whom immunotherapy is not feasible (vinflunine, docetaxel and paclitaxel) [6].

Erdafitinib is an alternative for platinum-refractory mUC with selected fibroblast growth factor receptor (*FGFR*) alterations (*FGFR2* mutations or *FGFR3* mutations/fusions) [7]. The antibody–drug conjugate enfortumab vedotin, targeting nectin-4, is currently the preferred option for patients with ChT and ICI-refractory disease [6,8].

Despite the recent advances and the incorporation of new drugs, the prognosis of mUC remains poor and there is a need of novel therapeutic strategies, including personalized therapies against new molecular targets. The identification of Her-2 as a potential candidate for targeted therapy in mUC has led to the development of monoclonal antibodies, tyrosine-kinase inhibitors, and antibody-drug conjugates with favorable results in clinical trials.

The purpose of this review is to summarize the biological rationale and the clinical results of anti-Her-2 therapy in patients with advanced urothelial cancer, both in monotherapy and in combination with other systemic treatments.

## 2. Her-2 in Urothelial Cancer: Biological Rationale

Her-2 is a protein encoded by the *ERBB2* oncogene, located at the long arm of chromosome 17 (17q12), and belonging to the family of epithelial growth factor receptors (HER1/EGFR, Her-2, HER3 and HER4) [9]. They are membrane tyrosine-kinase receptors (TKRs) involved in the activation of downstream intracellular signaling pathways such as the mitogen-activated protein kinase (MAPK) and the phosphoinositide 3-kinase (PI3K/Akt), that stimulate cell proliferation, differentiation and angiogenesis, amongst other essential events. The biological effects and downstream pathways activated by Her-2 are represented in Figure 1.

Her-2 overexpression has been well characterized in breast and gastric cancer, where it showed negative prognostic effects, probably driving the acquisition of multiple oncogenic and aggressive features in cancer cells [10]. Gene amplification is the primary mechanism for Her-2 overexpression in these tumors [11,12]. The patient selection is made by immunohistochemical (IHC) staining according to the modified DAKO criteria: negative (0/1+), equivocal (2+) and positive (3+), usually using chromogenic in situ hybridization (CISH) or fluorescence in situ hybridization (FISH) to determine *ERBB2* amplification in equivocal samples [12].

Urothelial cancer has the third highest rate of Her-2 overexpression, with mutations and amplification of *ERBB2* in 6–17% of samples, sometimes coexisting [13]. The most prevalent single-base substitution in UC is S310F (3.92% in bladder transitional cell carcinoma and 3.30% in upper tract tumors), followed by several low-incidence mutations (R103Q, D277Y, G292R, S310Y, F595L, R678Q, I767M, V777L, V842I, D933N) [14].

As it happens in breast and gastric tumors, Her-2 overexpression seems to be a negative prognosis factor in UC. A study of 150 patients with urothelial bladder cancer and lymph node metastases revealed that the rate of Her-2 amplifications—detected by FISH—was significantly higher in lymph node metastases (15.3%) than in primary tumors (8.7%) (*p* = 0.003), suggesting that Her-2 overexpression may play a pathogenic role in the lymphatic dissemination of UC [15].

A meta-analysis of nine studies with 2242 eligible patients with mUC demonstrated that Her-2 overexpression implied a worse disease-free survival (HR 1.68; 95% confidence interval [CI]: 1.33–2.14; *p* < 0.0001) and a poorer disease-specific survival (HR 2.00; 95% CI: 1.22–3.29; *p* = 0.006) [16]. Several other studies are consistent with the previous results and have associated Her-2 overexpression with a higher tumor stage—specially a higher lymph node involvement—and a worse disease-related survival [17,18].

Genome expression profiling has been used to classify urothelial bladder cancer into two main phenotypic and molecular subtypes—luminal and basal—with different markers that reflect the expression signature of intermediate/luminal and basal urothelial cell layers [18]. Tumors with luminal characteristics predominantly express the same cytokeratins as the normal urothelium (KRT18, KRT20) and markers of urothelial differentiation, such as uroplakins, and exhibit a stronger activation of the peroxisome proliferator activator receptor (PPAR). Basal-like tumors tend to express squamous features (KRT14, KRT15) and show a higher proliferation rate [14].

These two major phenotypes show distinct biological behaviors, the basal subtype being more aggressive but also more responsive to platinum-based ChT than the luminal subtype [19]. Kiss et al. [14] published one of the most complete analyses of Her-2 alterations at the DNA, RNA and protein level in 127 patients with MIBC, establishing the relevance of Her-2 as a tumor driver. According to their results, the rate of Her-2 alterations is higher in tumors with a luminal phenotype compared to the basal phenotype, suggesting that Her-2 activity is also associated with the molecular subtype.

Interestingly, this study found an imprecise correlation between *ERBB2* gene amplification and Her-2 overexpression—detected by IHC. FISH techniques revealed *ERBB2* amplification in 16/83 tumors (19.3%), which had a higher mRNA (*p* < 0.001) and Her-2 protein expression (*p* < 0.001), two parameters that were significantly related to each other (*p* < 0.001). However, Her-2 expression was negative by IHC in 5 of the 16 amplified samples (score 0/1) and equivocal in other 5 cases (score 2), suggesting that *ERBB2* amplification does not always lead to an increased protein expression. This might be explained by epigenetic factors, given that the highest rate of *ERBB2* methylation was observed in the *ERBB2* amplified cases with a lower mRNA expression (*p* < 0.001). On the other side, 19/67 non-amplified samples had a high mRNA expression and 13 of them had an IHC score of 3, meaning that gene amplification is not the only driver of Her-2 overexpression.

Another study in 61 patients with MIBC reported 10 cases of *ERBB2* amplification amongst 37 cases assessable by CISH (27%), with negative Her-2 expression (score 0) in 7/10, suggesting that gene amplification does not always match with the protein overexpression detected by IHC [20]. Besides this, the detection of gene amplification may vary depending on the technique, since FISH revealed amplification in only 5/42 cases (12%).

Therefore, the instruments for the selection of Her-2+ patients developed for breast and gastric cancer may not be optimal for UC patients. Assessing *ERBB2* status by FISH should always be considered, with the independence of the score obtained by IHC staining. As Her-2 is progressively positioned as a therapeutic target in mUC, further research is required to optimize the selection of patients most likely to respond to anti-Her-2 therapy.

## 3. Anti-Her-2 Drugs in UC: Clinical Results

Several Her-2-targeted therapies, both in monotherapy and in combination with other anti-Her-2 agents or other systemic treatments, have been evaluated in patients with mUC, with heterogeneous results. The main reported trials on this topic are shown in Table 1.

### 3.1. Anti-Her-2 Monoclonal Antibodies

Trastuzumab, the first humanized monoclonal antibody developed to target the Her-2 receptor, has been evaluated in combination with ChT in two phase II clinical trials. The first—published by Hussain et al. [21]—screened 109 patients with advanced UC for Her-2 overexpression, with a positive result by IHC or FISH in 57 patients (52.3%). A total of 44 Her-2+ patients were treated with trastuzumab, paclitaxel, carboplatin, and gemcitabine, with a median progression-free survival (mPFS) of 9.3 months and an overall response rate (ORR) of 70%, including 5 complete responses (CR) (11%) and 26 partial responses (PR) (59%). The toxicity of the combination was significant, with G3/G4 adverse events in 41 patients (93%) and 2 toxic deaths (5%). This study showed that Her-2 positive patients had a significantly higher median number of metastatic sites (2 vs. 1; *p* = 0.014) and a higher incidence of two or more metastatic sites (51% vs. 31%; *p* = 0.051) compared with Her-2 negative patients, which is coherent with the previous data regarding the unfavorable prognosis implications of Her-2 overexpression.

Oudard et al. published a second phase II randomized trial [22] evaluating ChT (platinum + gemcitabine) with or without trastuzumab in the first line setting for Her-2 positive advanced or metastatic UC. Among 563 screened patients, 75 (13.3%) were Her-2 positive (IHC 2+/3+ and FISH+) and 61 met all eligibility criteria. No significant differences were detected between both groups in mPFS (10.2 vs. 8.2 months; *p* = 0.689), ORR (65.5% vs. 53.2%; *p* = 0.39) and median overall survival (mOS) (15.7 vs. 14.1 months; *p* = 0.684). The unexpectedly low incidence of Her-2 overexpression was discussed as a possible cause for the lack of statistical power of this study. The addition of trastuzumab to ChT did not have any impact on the quality-of-life scales.

More recently, the phase II basket study MyPathway [23] has evaluated the dual Her-2 blockade with trastuzumab plus pertuzumab in a large tissue-agnostic cohort of patients with refractory Her-2 positive advanced solid tumors. The study included 22 mUC (18 Her-2 amplification, 3 Her-2 amplification + mutation, 1 Her-2 amplification + overexpression) and reported an ORR of 15.8% (lower than the ORR reported in salivary glands [63.6%], pancreatic [33.3%], colorectal [30.9%], biliary tract [25.7%] and non-small-cell lung cancer [23.8%], but higher than in ovarian [10.0%] and uterine [6.3%] tumors). *KRAS* mutation was identified as a predictor of non-response to Her-2 blockade. As expected, this scheme showed a much better safety profile than the combination of trastuzumab with ChT.

### 3.2. Anti-Her-2 Tyrosine Kinase Inhibitors

While monoclonal antibodies are directed at the extracellular domain of the receptor, tyrosine-kinase inhibitors (TKIs) are low-molecular-weight molecules that act as competitive inhibitors of the tyrosine-kinase enzymes located in the intracellular part of the receptor, avoiding the phosphorylation of tyrosine residues that unleash the activation of downstream signaling pathways. The main TKIs against Her-2 are lapatinib, afatinib and neratinib, and have been studied in monotherapy and in combination with conventional ChT.

Lapatinib was first evaluated in two phase II clinical trials for patients with refractory mUC. Galsky et al. [24] reported a total of 32 patients with Her-2 amplified solid tumors treated with lapatinib, including nine mUC (28%), three of them reaching stable disease, without any objective response (disease control rate [DCR]: 33%). Wülfing et al. [25] studied lapatinib as a second-line therapy in 59 patients with advanced or metastatic UC, not selected by Her-2 status. High expression (IHC 2+/3+) of EGFR and Her-2 was detected in 30 patients (52%) and 25 patients (44%), respectively. Disease control was achieved in a total of 19 patients (32%) (1 PR and 18 SD), and it was found to be correlated with the overexpression of EGFR and/or Her-2, which was detected in the 89% of patients that achieved clinical benefit. Patients with EGFR/Her-2 overexpression treated with lapatinib had a significantly higher median OS than non-expressing patients (30.3 weeks vs. 10.6 weeks; *p* = 0.0001).

More recently, Powles et al. [26] studied lapatinib as first-line maintenance treatment in a phase III clinical trial. A total of 232 patients with Her-1/Her-2 positive mUC who did not have progressive disease during initial ChT (cisplatin-based ChT in 62.1%) were randomly assigned to maintenance with lapatinib or placebo. Among the 116 patients receiving lapatinib, 53 were Her-1+ (45.7%), 21 were Her-2+ (18.1%) and 42 were Her-1 and Her-2 positive (36.3%). No significant differences were observed between both groups in terms of median PFS (4.5 m vs. 5.1 m; *p* = 0.63) and OS (12.6 m vs. 12.0 m; *p* = 0.80), showing that the clinical outcome of Her-2 positive mUC is not improved by the addition of lapatinib to standard ChT.

Two studies have combined lapatinib with conventional ChT in different clinical sceneries, with disappointing results. Cerbone et al. [27] conducted a phase I clinical trial of cisplatin plus gemcitabine and lapatinib as first line therapy for patients with mUC. A total of 17 patients were enrolled, not selected by Her-2 status (IHC study was performed in 16 samples, revealing one 3+, four 2+, eight negative and three unknown results). Among the 17 patients, 10 responded to treatment (1 CR, 9 CR) (ORR 58.8%) and 4 patients had SD. Tang et al. [28] evaluated the combination of docetaxel and lapatinib in a phase II clinical trial that included 15 patients with refractory mUC. They achieved 1 CR (8%) and 4 SD (31%), with a median PFS of 2.0 months and a median OS of 6.3 months. The data from these studies suggest a lack of benefit of lapatinib for UC, alone or in combination with conventional ChT, unless new markers of response were identified.

Afatinib is another ErbB-family irreversible TKI with modest results in UC, according to the results of three phase II clinical trials. Choudhury et al. [29] studied the efficacy of afatinib in 23 mUC patients, achieving clinical benefit in 5 cases (21.7%) (1 PR, 4 SD). Interestingly, among the 21 tumor samples analyzed, 6 had Her-2 and/or ErbB3 alterations and the 5 patients with clinical benefit belonged to this group, whereas 15 out of 15 patients without ErbB alterations had progressive disease as best response. The median PFS was significantly higher in patients with Her-2/ErbB3 alterations (6.6 months vs. 1.4 months; *p* < 0.001), confirming their response-predicting value. The phase II LUX-Bladder 1 trial [30] evaluated afatinib in 42 mUC patients with ErbB alterations, divided into two cohorts: 34 patients with Her-2/Her-3 amplification or mutation (cohort A) and 8 patients with EGFR amplification (cohort B). A total of 2 PR and 15 SD were observed among the patients from cohort A (DCR: 50%), with a median PFS of 9.8 weeks and 4 patients (11.8%) free of progression at 6 months. The phase II NCI-MATCH EAY131 basket trial [31] studied afatinib in 40 patients with refractory solid tumors with Her-2 activating mutations, including 4 patients with mUC, among which no objective response was observed.

The basket trial SUMMIT [32] evaluated the pan-HER kinase inhibitor neratinib in a wide range of advanced solid tumors with somatic mutations of Her-2 and ErbB3. From a total of 141 patients, 18 mUC cases were included (12.8%) (16 Her-2 mutant and 2 ErbB3 mutant). As with other solid tumors, S310 was the most frequent Her-2 mutation, followed by V777 and exon 20 insertion. Among patients with mUC, 3 SD were achieved (18.8%), though no objective response was observed. The SUMMIT trial concluded that neratinib had the greatest activity in breast, cervical and biliary cancers, as well as in tumors that contained kinase-domain missense mutations.

### 3.3. Antibody-Drug Conjugates: A Novel Promising Strategy in Anti-Her-2 Therapy

Antibody-drug conjugates (ADCs) are complex molecules based on the combination of a monoclonal antibody, highly selective for a tumor-associated antigen, and an ultra-toxic payload that induces the death of the target cell after being internalized and released. Both parts are connected by a linker component that, in ideal conditions, should be stable in blood circulation but cleavable at the target site. Technological upgrades such as the use of humanized antibodies to reduce immunogenicity, the increasing specificity of antibodies, the better linker stability, and the higher potency of the cytotoxic payload, have played a key role in the progressive improvement of ADCs’ quality and results.

ADCs with different targets are changing the therapeutic landscape in UC. Anti-nectin 4 ADC enfortumab vedotin has been recently approved for refractory mUC, after achieving an ORR of 40.6% in a pivotal phase III clinical trial [38]. Promising results have been obtained in phase I/II trials for mUC patients with anti-Trop2 sacituzumab govitecan (ORR 27% [39]). ADCs are also being evaluated in the context of non-muscle invasive bladder cancer, with the anti-epithelial cell adhesion molecule (EpCAM) ADC oportuzumab monatox achieving high rates of complete pathological response in a phase II clinical trial [40]. In this context, ADCs have also emerged as a promising new strategy for Her-2 targeted therapy.

Trastuzumab emtansine (TDM-1) was the first developed anti-Her-2 ADC, combining an anti-Her-2 monoclonal antibody with an anti-microtubule agent (DM-1), covalently linked via a stable linker to release the cytotoxic drug in Her-2-expressing cells. The growth of platin-resistant tumors in preclinical models was significantly more inhibited by TDM-1, via induction of apoptosis, in comparison with control IgG or trastuzumab [41]. A multi-histology basket trial of TDM-1 recruited a total of 58 patients with Her-2 amplified tumors, showing a global ORR of 26%, though—despite the promising preclinical results—no response was observed in patients with mUC [42]. More recently, a multicenter exploratory phase II clinical trial (KAMELEON) has been designed to assess TDM-1 in Her-2 overexpressing solid tumors, enrolling 13 patients in the cohort for urothelial bladder cancer. Preliminary results show 38.5% partial responses and a median PFS of 2.2 months, with 53.8% adverse events greater than grade 3 [43]. An ongoing phase II trial is studying the combination of TDM-1 with TKI tucatinib (NC04632992).

Trastuzumab duocarmazine is an ADC that contains trastuzumab covalently bound to a duocarmycin prodrug that acts as a cytotoxic agent. Banerji et al. [33] published a phase I dose-expansion basket clinical trial with trastuzumab duocarmazine in 146 patients with Her-2-expressing solid tumors, including 16 patients with mUC. An ORR of 25% was achieved, with a DCR of 94% and a median PFS of 4.0 months. The combination of trastuzumab duocarmazine with paclitaxel is being evaluated by an ongoing phase I trial (NCT04602117).

Disitamab vedotin is a novel humanized anti-Her-2 ADC that is comprised of monomethyl auristatin E (MMAE) coupling hertuzumab, a new-generation anti-Her-2 monoclonal antibody that targets different epitopes of the Her-2 receptor and binds to it with a higher molecular affinity than trastuzumab. In addition, MMAE released by disitamab vedotin has shown a better membrane penetration that seems to induce remarkable bystander effects, which may enhance its efficacy against solid tumors [44]. Promising results and a manageable safety profile have been reported in Her-2 positive UC. Sheng et al. [34] presented a combined analysis of two multi-center phase II clinical trials (RC48-C005 [45], RC48-C009 [46]) evaluating disitamab vedotin in a total of 107 patients with advanced or metastatic Her-2 positive UC (IHC 2+/3+) after progression to platinum-based ChT. The ORR was 50.5%, with a median PFS of 5.9 months and a median OS of 14.2 months. The highest response rate was achieved in the subgroup of IHC 3+ or IHC2+/FISH+ patients (62.2%). The treatment tolerance was adequate, with a low rate of G3/G4 adverse events.

Xu et al. [35] have recently presented a phase II clinical trial evaluating disitamab vedotin in 19 patients with Her-2 negative mUC (6 patients IHC 0 and 13 patients IHC 1+), achieving a median PFS of 5.5 months, a global ORR of 26.5% and a DCR of 94.7%, with 5 PR (all of them in patients with IHC 1+) and 13 SD. The treatment was well tolerated, with a 15.8% of G3 adverse events. The authors concluded that ADCs could be a treatment option for patients with Her-2 negative mUC, especially those with IHC 1+.

Trastuzumab deruxtecan (T-DXd) is an ADC that combines the monoclonal anti-Her-2 antibody with a topoisomerase I inhibitor. T-DXd is specifically designed to improve the characteristics of other anti-Her-2 ADCs, showing a higher drug-to-antibody ratio (DAR 8 vs. 3–4) and carrying a payload with a short half-life that increases its cytotoxic effect but minimizes the off-target toxicity in healthy cells [47]. Besides this, the T-DXd payload has a high membrane permeability, resulting in an increased bystander effect, which allows a cytotoxic effect on tumor cells close to targeted cells with independence of their Her-2 expression levels. This might explain the success of T-DXd in Her-2 low advanced breast cancer [48] and the higher sensitiveness to T-DXd observed in tumors harboring heterogeneous Her-2 expression, such as colorectal and non-small-cell lung cancer, in comparison with other anti-Her-2 agents [49].

Several preclinical studies support the favorable pharmacologic activity of T-DXd both in Her-2 low expressing tumors and in tumors expressing Her-2 protein in the absence of Her-2 amplification, thus potentially broadening the population of patients that might benefit from Her-2 targeted therapy [50]. Two ongoing phase II basket trials (NCT04482309/DESTINY-PanTumor02 and NCT04639219) are currently evaluating T-DXd in Her-2-expressing metastatic solid tumors, including cohorts for patients with mUC.

The combination of ADCs + immune checkpoint inhibitors is an innovative strategy in treatment of advanced urothelial carcinoma and could show even better results than the use of ADCs alone. Preclinical studies suggest that ADCs linked to microtubule-depolymerizing agents can induce immunogenic cell death and enhance anti-tumor immune activity, by stimulating the population of intratumoral CD45+ dendritic cells [51,52]. Furthermore, the ADCs seem to favor the membrane expression of immune-activating coreceptors such as CD86, as well as the expression of PD-L1 and major histocompatibility complex (MHC) class I [52]. This implies that anti-Her-2 ADCs might activate immune response via mechanisms other than their cytotoxic effects, rising an exciting rationale for their combination with immunotherapy. One very relevant point in this field will be the identification of patient-specific biomarkers able to predict, in the preclinical setting, the response of combination strategies of ADCs and immunotherapy. The implication of a number of these biomarkers has been recently envisioned in bladder cancer [53].

Zhou et al. [36] have published preliminary results of 14 patients with Her-2 positive mUC enrolled in a phase Ib/II trial of disitamab vedotin combined with anti-PD-1 monoclonal antibody toripalimab, achieving an impressive ORR of 80% and a DCR of 90% (8 PR and 1 SD among 10 patients assessable for response). All responsive patients showed durable responses regardless of Her-2 expression, and the safety profile was favorable, with mostly grade 1–2 adverse events. Follow-up continues for the evaluation of PFS and OS.

Galsky et al. [37] have communicated a primary analysis from a phase Ib/II trial combining T-DXd and nivolumab in 34 patients with Her-2-expressing mUC, achieving an ORR of 36.7% (4 CR and 7 PR among 30 patients with IHC 2+/3+, and 2 PR among 4 patients with IHC 1+). The median PFS was 6.9 months, and the median duration of response was 13.1 months. The number of responders was higher among patients with a tumor mutation burden (TMB) ≥10 mutations/Mb and a positive PD-L1 expression (CPS ≥1). The rate of adverse events reported with this combination was significant, with a 73.5% of G3/4 AEs (44.1% related to T-DXd and 26.5% related to nivolumab) and one toxic death due to immune-related pneumonitis.

The combination of TDM-1 plus atezolizumab is under evaluation by the phase II clinical trial MyTACTIC (NCT04632992) in patients with refractory Her-2 positive mUC. An ongoing phase I clinical trial (NCT03523572) is studying the combination of T-DXd and nivolumab in patients with Her-2 positive breast and bladder cancer.

Another emerging strategy is the development of peptide-drug conjugates (PDCs). PDCs are investigational agents that combine a cytotoxic molecule with a low-molecular weight peptide through a biodegradable linker. These small peptides can be designed with a specific sequence of amino acids to confer desirable pharmacokinetic properties upon them, such as the possibility of self-assembling; temporarily limit the drug’s bioactivity to minimize its premature liberation; and selectively bind to or within the surface of tumor or vascular endothelial cells [54]. For example, basic research works have identified an hexapeptide (KCCYSL) commonly involved in a population of phages with a high affinity against the extracellular domain of Her-2 [55]. This knowledge could be used to synthetize new high-affinity peptides with selective delivery into malignant Her-2-positive cells, therefore allowing the use of innovative cytotoxic payloads and enhancing their anti-tumor activity.

Early evidence suggests that several mechanisms can contribute to the development of resistance to ADCs, including the lack of antibody attachment, the loss of the payload cytotoxic effect and the alteration of the ADC process of internalization [56]. Much research is still needed to properly understand the resistance mechanisms and to identify predictive biomarkers of response that help us optimize the use of ADCs in urothelial cancer.

## 4. Conclusions

A significant percentage of urothelial tumors (10–20%) show Her-2 overexpression or amplification, which has been associated with an aggressive biological behavior and a poor clinical prognosis. Her-2 targeted therapy has emerged as a new therapeutic approach for advanced or metastatic UC. The use of anti-Her-2 monoclonal antibodies (trastuzumab, pertuzumab) and tyrosine-kinase inhibitors (lapatinib, afatinib, neratinib) in monotherapy—for patients with ChT-refractory tumors—has shown modest results in phase II clinical trials, with overall response rates lower than 10%. Their combination with conventional ChT in the first line setting does not seem to significantly improve the benefit of standard treatment.

Antibody–drug conjugates targeting a wide variety of molecules, including Her-2, have emerged as an interesting strategy for the treatment of several tumors. Anti-Her-2 ADCs such as trastuzumab emtansine and trastuzumab duocarmazine have achieved positive results in multi-tumor basket clinical trials, with response rates over 25% in pre-treated patients with mUC. Disitamab vedotin has shown impressive results in a phase II study, with a response rate over 60% in IHC 3+ or FISH+ urothelial tumors. Furthermore, interesting results have been presented in Her-2 negative tumors. With an improved pharmacokinetic profile, trastuzumab deruxtecan is also a promising ADC for patients with Her-2 positive and low-expressing tumors, currently under evaluation by several basket trials.

The combination of disitamab vedotin with anti-PD-1 toripalimab has obtained outstanding preliminary results, with a rate of objective responses that might ber 80%, suggesting a synergistic effect of ADCs and checkpoint inhibitors due to their capacity to elicit anti-tumor immunity. Further research is required to understand the mechanisms of resistance to ADCs and develop highly specific peptide–drug conjugates that might improve the clinical benefit of Her-2 targeted therapy in urothelial cancer.

## Figures and Tables

**Figure 1 ijms-23-12659-f001:**
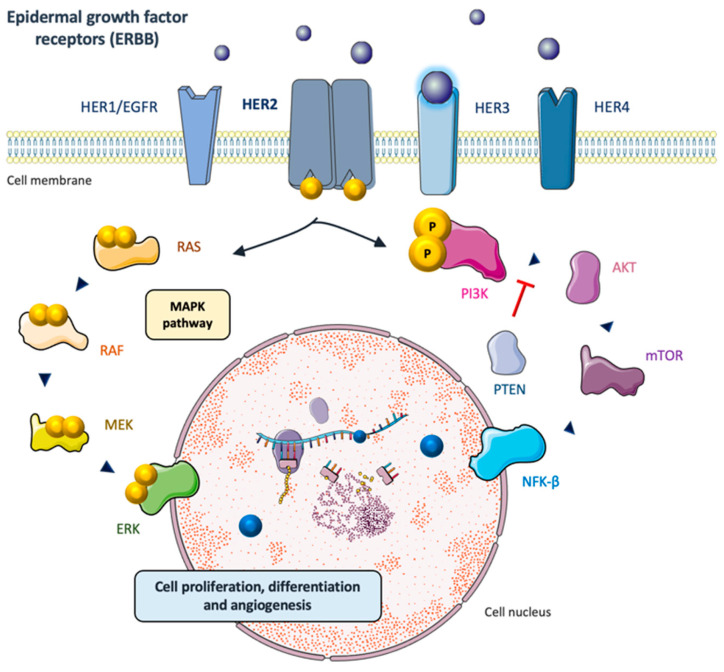
Main downstream oncogenic pathways unleashed by Her-2 activation. Unlike other receptors from ERBB family, Her-2 does not have an extracellular ligand and its heterodimerization is promoted by the ligand-binding of the other three receptors (HER1/EGFR, HER3, HER4). Its activation unleashes the phosphorylation of intracytoplasmic tyrosine substrates and the initiation of downstream transduction pathways with pro-oncogenic effects.

**Table 1 ijms-23-12659-t001:** **Clinical trials of anti-HER-2 therapy for urothelial cancer with published results**. mUC: metastatic urothelial cancer; AEs: adverse events; GI: gastrointestinal; CV: cardiovascular; mPFS: median progression-free survival; mOS: median overall survival; mDR: median duration of response; CR: complete response; PR: partial response; SD: stable disease; DCR: disease control rate (PR+SD); ORR: overall response rate; m: months; w: weeks; AST: aspartate aminotransferase; ALT: alanine aminotransferase; GGT: gamma-glutamyl transferase; PPS: palmar-plantar syndrome; wt: wild type; IQR: inter-quartile range; LVEF: left ventricular ejection fraction; NA: not available; NE: not estimable. (*) Study without published results; data available at https://clinicaltrials.gov/ct2/show/results/NCT02999672 (accessed on 17 October 2022).

Clinical Trial (Phase)	Agents	Context	*N*	Her-2 Selection	Age Range (Years)	Outcomes	G3/G4 Adverse Events
Hussain et al. (phase II) [21]	Trastuzumab + carboplatin + paclitaxel + gemcitabine	1st line (mUC)	44	Yes	37−89	ORR: 70% (95% CI: 55−83%); CR: 5 (11%); PR: 26 (59%); SD: 5 (11%); DCR: 36 (81%) mPFS: 9.3 m (95% CI 6.7−10.2); mOS: 14.1 m (95% CI 11.5−17.1); mDR: 7.1 m (95% CI 4.8−8.0)	Neutropenia (86%), thrombopenia (70%), anemia (45%), GI AEs (28%), metabolic alterations (16%), neuropathy (14%), CV AEs (14%), fatigue (14%), renal failure (14%), neutropenic fever (13%), hemorrhage (11%), dyspnea (9%), elevated ALT/AST (7%)
Ourdard et al. (phase II, randomized) [22]	Platinum + gemcitabine +/- trastuzumab	1st line (mUC)	61	Yes	45−80	ORR: 19/29 (65.5%) vs 17/32 (53.2%); CR: 20.7% vs 21.9%; PR: 44.8% vs 31.3%; mPFS: 10.2 m (95% CI 4.3−13.4) vs 8.2 m (95% CI 4.6−10.6); mOS: 15.7 m (95% CI 12.2−23.6) vs 14.1 m (95% CI 9.3−28.0)	Neutropenia (75.9% vs 67.7%), thrombopenia (48.3% vs 38.7%), anemia (41.4% vs 38.3%), GI AEs (3.4% vs 3.1%), dyspnea (0% vs 3.1%), neutropenic fever (3.4% vs 9.7%)
Meric-Bernstam et al. (MyPathway) (basket phase II) [23]	Trastuzumab + pertuzumab	Refractory solid tumors	258 (22 mUC)	Yes	23−87	All tumors: ORR: 60/258 (23.3%) (95% CI 18.2−28.9%) (5 CR, 55 PR); DCR: 115/258 (44.6%) (95% CI 38.4−50.9%); mPFS: 2.8 m (95% CI 2.7−4.0); mDR: 7.9 m (95% CI 6.2−9.3); mOS: 10.9 m (95% CI 9.2−13.8); mUC (*KRAS* wt): ORR: 15.8%	Anemia (1.6%), diarrhea (1.6%), infusion-related reaction (1.6%)
Galsky et al. (basket phase II) [24]	Lapatinib	Refractory solid tumors	32 (9 mUC)	Yes	39−85	All Her-2+ tumors: CR: 1/32 (3%); PR: 0/32; SD: 9/32 (28%); mPFS: 78 d (95% CI: 42–118) mUC: ORR: 0%; DCR: 33% (SD)	G3 AEs occurring in >5%: nausea and diarrhea (G3 AEs in <5%: not specified)
Wülfing et al. (phase II) [25]	Lapatinib	2nd line (mUC)	59	No (25/59 Her-2+)	41−78	ORR: 1 (2%) (95% CI 0.04–9.09); PR: 1 (2%); SD: 18 (31%); DCR: 19 (32%) (5/27 Her-2 0/1+ [19%]; 14/30 Her-2 2+/3+ [47%]); mPFS (EGFR/Her-2+): 9.0 w (95% CI 8.1–15.0); mPFS (EGFR/Her-2-): 7.9 w (95% CI 4.3−8.3); mOS (EGFR/Her-2+): 30.3 w (95% CI 17.6−49.9); mOS (EGFR/Her-2-): 10.6 w (95% CI 5.0−17.6)	Vomiting (7%), diarrhea (3%), dehydration (3%), hyponatremia (3%)
Powles et al. (phase III, randomized) [26]	Lapatinib	Maintenance after 1st line ChT	232	Yes (Her-1/2)	64.2−77.1 (IQR)	mPFS: 4.5 m (95% CI 2.8−5.4) (lapatinib) vs 5.1 m (95% CI 3.0−5.8) (placebo) (HR 1.07; 95% CI 0.81−1.43) (*p* = 0.63); mOS: 12.6 m (95% CI 9.0−16.2) (lapatinib) vs 12.0 m (95% CI 10.5−14.9) (placebo) (HR 0.96; 95% CI 0.70−1.31) (*p* = 0.82).	Pain (10.3% vs 5.1%), diarrhea (6.2% vs 1.0%), infection (5.2% vs 4.0%), fatigue (4.1% vs 1.0%), vomiting (3.1% vs 1.0%), constipation (2.1% vs 1.0%), rash (2.1% vs 0.0%), nausea (1.0% vs 1.0%)
Cerbone et al. (phase I) [27]	Cisplatin + gemcitabine + lapatinib	1st line (mUC)	17	No (5/17 Her-2+)	49.6−76.9	ORR: 10/17 (58.8%); DCR: 14/17 (82.4%) (1 CR, 9 PR, 4 SD)	Neutropenia (70.6%), lymphopenia (23.5%), thrombocytopenia (41.2%), neutropenic fever (11.8%), renal failure (11.8%), pulmonary side effects (5.9%), nausea (5.9%)
Tang et al. (phase II) [28]	Docetaxel + lapatinib	Refractory (mUC)	15	No (not specified)	NA	ORR: 1/15 (8%) (1 CR); PR: 0/15; SD: 4/15 (31%); mPFS: 2.0 m (95% CI 1.3–6.6); mOS: 6.3 m (95% CI 2.2−12.7)	Diarrhea (33%), vomiting (26.7%), nausea (26.7%), fatigue (6.7%)
Choudhury et al. (phase II) [29]	Afatinib	Refractory (mUC)	25	No (6/25 Her-2/3+)	36–82	ORR Her-2/3+: 5/6 (83.3%) / Her-2/3-: 0/15 (0%) mPFS: 6.6 m for Her-2/3+ vs 1.4 m for Her-2/3-	Fatigue (13%), diarrhea (8.7%), acneiform rash (8.7%), vomiting (8.7%), acute renal failure (4.3%), chronic kidney disease (4.3%), PPS (4.3%), cough (4.3%), pleural effusion (4.3%)
Font et al. (LUX-Bladder 1) (phase II) [30]	Afatinib	Refractory (mUC)	42	Yes	NA	Cohort A (Her-2/3 mutation/amplification): ORR: 2/34 (5.9%); DCR: 17/34 (50%) (2 PR, 15 SD); mDR: 22.7 w (95% CI 15.1−36.1); PFS at 6 m: 4/34 (11.8%); mPFS: 9.8 w (95% CI 7.9−16.0); mOS: 30.1 m (95% CI 17.4−47.0)	Infection (26.2%), connective tissue disorders (23.8%), renal failure (7.1%), nausea (4.8%), pyrexia (4.8%), asthenia (2.4%), pain (2.4%), anemia (2.4%), acute coronary syndrome (2.4%), cardiac failure (2.4%), dysphagia (2.4%), intestinal obstruction (2.4%), cachexia (2.4%), pulmonary embolism (2.4%), ischaemic stroke (2.4%), basal cell carcinoma (2.4%)
Bedard et al. (basket phase II) [31]	Afatinib	Refractory solid tumors	40 (4 mUC)	Yes	NA	All tumors: ORR: 1/40 (2.7%) (90% CI 0.14−12.2) (1 PR); PFS at 6 m: 12.0% (90% CI 5.6−25.8) mUC: ORR: 0/4 (0%)	Not specified, most AEs G1/2
Hyman et al. (SUMMIT) (basket phase II) [32]	Neratinib	Refractory solid tumors	141 (18 mUC)	Yes	30−83	All Her-2+ tumors: ORR at week 8: 13/141 (9.2%); DCR: 39/141 (27.7%)	Diarrhea (22.0%), anemia (7.1%), dehydration (5.7%), abdominal pain (5.0%), nausea (4.2%), fatigue (3.5%), elevated AST (3.5%), dyspnea (3.5%), constipation (1.4%), hyporexia (0.7%), asthenia (0.7%)
						Her-2+ mUC: ORR: 0%; DCR: 3/16 (18.8%) (95% CI 4.0−45.6)	
KAMELEON (*)	Trastuzumab emtansine (TDM-1)	Refractory solid tumors	20 (13 mUC)	Yes	NA	CR: 0/13; PR: 5/13 (38.5%); mPFS 2.2 m (95% CI 1.18−4.30); mOS 7.03 m (95% CI 3.75-NE)	G3: 30.8%; G4: 0%; G5: 23.1% (not specified)
Banerji et al. (basket phase I) [33]	Trastuzumab duocarmazine	Refractory solid tumors	146 (16 mUC)	Yes	47–71	ORR: 4/16 (25%) (95% CI 7.3−52.4%) DCR: 15/16 (93.8%) (4 PR, 11 SD) mPFS: 4.0 m (95% CI 1.3-NE)	Neutropenia (6%), conjunctivitis (4%), fatigue (3%), lymphopenia (3%), keratitis (2%), thrombopenia (2%), pleural effusion (2%), LVEF decreased (2%) dry eye (1%), decreased appetite (1%), vomiting (1%), infusion-related reaction (1%), vision blurred (1%), hemoptysis (1%), diarrhea (1%), elevated AST (1%), hematuria (1%), dyspnea (1%), anemia (1%), mouth ulceration (1%), rash maculo-papular (1%), elevated ALT (1%), leucopenia (1%), retinal haemorrhage (1%),
Sheng et al. (RC48C005/RC48C009) [34]	Disitamab vedotin	Refractory (mUC)	107	Yes	40−79	Overall ORR: 50.5% (95% CI 40.6−60.3%); ORR in IHC 3+ or 2+/FISH+: 62.2%; mPFS: 5.9 m (95% CI 4.2−7.2); mOS: 14.2 m (95% CI 9.7−18.8)	Hypoesthesia (15.0%), neutropenia (12.1%), elevated GGT (5.6%)
Xu et al. (phase II) [35]	Disitamab vedotin	Her-2 negative refractory (mUC)	19	Yes	NA	Overall ORR: 5/19 (26.3%) (95% CI 9.1−51.2%) ORR IHC 0/6; ORR IHC 1+: 5/13 (38.5%) DCR: 18/19 (94.7%) (95% CI 74.0−99.9%) mPFS: 5.5 m (95% CI 3.9−6.8)	Neutropenia (10.5%), white blood cell count decreased (5.3%)
Zhou et al. (phase Ib/II) [36]	Disitamab vedotin + toripalimab	Refractory (mUC)	14	No (not specified)	52−76	ORR: 8/10 assessable patients (80%); DCR: 9/10 (90%) (8 PR, 1 SD). Follow-up continues for PFS/OS	Intestinal obstruction attributed to treatment in 1/14 (7%)
Galsky et al. (phase Ib/II) [37]	Trastuzumab deruxtecan + nivolumab	Refractory (mUC)	34	Yes	41−80	Overall ORR: 13/34 (38.2%). ORR in IHC 2+/3+: 11/30 (36.7%) (95% CI 19.9−56.1%) (4 CR, 7 PR) mDR: 13.1 m (95% CI 4.1−NE); mPFS: 6.9 m (95% CI 2.7−14.4); mOS: 11.0 m (95% CI 7.2−NE)	G3/G4 AEs >5%: AST increase, fatigue, thrombopenia, pneumonitis

## Data Availability

Not applicable.

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
