# Peer review of "Her-2 Targeted Therapy in Advanced Urothelial Cancer: From Monoclonal Antibodies to Antibody-Drug Conjugates"

_ijms, 2022, doi:10.3390/ijms232012659_

Round 1
Reviewer 1 Report
This manuscript is well written and offers interesting perspectives in the field of Her-2 targeted therapy in advanced urothelial cancer. However, it needs a number of changes before being accepted for publication in the International Journal of Molecular Sciences. The changes needed are detailed in the text below.
TITLE: The most important concept transmitted by this review, which is also its main strength, is that antibody-drug conjugated are bringing some hope to the grim field of systemic treatment of metastatic urothelial carcinoma. This is the real novel concept underlined by this paper.
I think therefore that this fact should be clearly stated in the title of the paper, which is at the moment a bit generic and not very attention-catching. A proposed change for the title is:
ANTIBODY-DRUG CONJUGATES AS A NOVEL PROMISING STRATEGY IN HER-2 TARGETED THERAPY OF ADVANCED UROTHELIAN CANCER: PRELIMINARY DATA AND FUTURE PERSPECTIVES
ABSTRACT:
Line 9: metastatic urothelial cancer, associated with a poor prognosis, is a major cause of cancer- related death…
Line 11-16: Re-phrase, for example: The human epithelial growth factor receptor 2 (Her-2) has been identified as a new therapeutic target in Medical Oncology. However, despite the encouraging results in breast and gastric cancers, clinical trials with anti-Her-2 monoclonal antibodies and tyrosine-kinase inhibitors have shown limited efficacy of this strategy in urothelial tumors. Notably, more favourable data have been recently shown by the use of antibody-drug conjugates, currently emerging as a novel promising approach for Her-2 targeted therapy in advanced urothelial cancer.
I think this is much more incisive as an abstract.
INTRODUCTION:
General comment:
Be careful with adjectives: you use in the manuscript: OUTSTANDING, INTERESTING, IMPRESSIVE, and other overstating adjectives. It is best to understate concepts and just report the scientific evidence. Line 41, for example, you say “interesting”, but then in the abstract (line 16) you say “outstanding” for the same thing. It is quite a different concept! Just say “favourable results” (Introduction, line 41). Moreover, erase” outstanding results” in the abstract (see text above).
Page 2, Line 60: Her-2 overexpression has been well characterised in breast and gastric cancer, where it showed negative prognostic effects, probably driving the acquisition of multiple oncogenic and aggressive features in cancer cells.
Page 3, line 73-78: re-phrase, unclear
Page 3, line 121: ANTI-HER-2 DRUGS IN UC: CLINICAL RESULTS
Page 4, line 124: The main reported trials on this topic are shown in Table 1
Page 5, in the sub-title (line 222): Antibody-drug conjugates: a novel promising strategy in anti-Her-2 therapy of advanced urothelial cancers
Page 7: line 297-305: this is very novel paragraph, absolutely interesting in terms of future perspectives of what you are reviewing in this manuscript. You need to re-phrase it better and add a citation which refers to previously work published on what you are stating. “The combination of immune checkpoint inhibitors + ADCs is a really innovative strategy in treatment of advanced urothelial carcinoma, and could show even better results than the use of ADCs alone. One very relevant point in this field will be the identification of patient-specific biomarkers able to predict, in the preclinical setting, response of combination strategies of ADCs and immunotherapy. The implication of a number of these biomarkers has been recently envisioned in bladder cancer” (ref. Mancini M., et al, , 2021, DOI: 10.3390/cancers13236016).
Page 7, line 327-331 re-phrase more clearly.
Page 8, line 333-337: re-phrase more clearly.
Reviewer 2 Report
Authors provide a review on Her-2 targeted therapy in advanced urothelial cancer (UC). The authors need to be congratulated on their survey. It covers relevant and interesting data on the current therapeutic landscape of available medication for this disease. The abstract is concise and provides sufficient information on the content of the survey. The introduction is reasonable and provides an introduction to the topic of the review. Authors cover major possible mechanisms of Her-2 in a reasonable and concise manner in the main body of the manuscript. However, and as this is a survey, please comment on some minor issues. In the introduction you state “Urothelial bladder cancer (UC) is among the top ten most frequent cancer types in the world and accounts for around 3% of new malignant diagnosis […]”. This is correct. Furthermore, at time of diagnosis one fourth of patients already presents with muscle invasive bladder cancer (MIBC). Although the majority of patients does not present with muscle invasive tumor stages, up to 50 % with high risk non-muscle invasive bladder cancer (NMBC) progress to invasive disease. However, RC remains the gold standard of treatment for non-metastasized muscle invasive bladder cancer (MIBC), even for patients >85 years (PMID: 32871580). Unfortunately, half of patients metastasize after curative RC (PMID: 11157016). With this in mind, it becomes obvious that the patient population requiring systemic treatment for mUC is old and vulnerable, and may not be eligible for platinum-based chemotherapy, and thus, require alternatives. I think, the introduction would profit from adding this.
Author Response
Thank you so much for your favorable evaluation. The recommended changes have been incorporated to the main text as suggested (see page 1, lines 24-28).